# Osteoconductive Effect of a Nanocomposite Membrane Treated with UV Radiation

**DOI:** 10.3390/polym14020289

**Published:** 2022-01-11

**Authors:** Yusser Olguín, Soledad Acuna-Mendoza, Carolina Otero, Cristian A. Acevedo, Cristian Covarrubias

**Affiliations:** 1Centro Científico Tecnológico de Valparaíso CCTVaL, Universidad Técnica Federico Santa María, Valparaíso 2390123, Chile; cristian.acevedo@usm.cl; 2Department of Oral Pathology and Medicine, Faculty of Dentistry, University of Chile, Santiago 8380453, Chile; sam@odontologia.uchile.cl; 3Escuela de Química y Farmacia, Facultad de Medicina, Universidad Andres Bello, Santiago 8370149, Chile; maria.otero@unab.cl; 4Centro de Biotecnología, Universidad Técnica Federico Santa María, Valparaíso 2390123, Chile; 5Departamento de Física, Universidad Técnica Federico Santa María, Valparaíso 2390123, Chile; 6Laboratory of Nanobiomaterials, Institute for Research in Dental Sciences, Faculty of Dentistry, University of Chile, Santiago 8380453, Chile; ccovarrubias@odontologia.uchile.cl

**Keywords:** bone regeneration, uv-treatment, biomaterials, calvarial defect

## Abstract

Modulation of the bio-regenerative characteristics of materials is an indispensable requirement in tissue engineering. Particularly, in bone tissue engineering, the promotion of the osteoconductive phenomenon determines the elemental property of a material be used therapeutically. In addition to the chemical qualities of the constituent materials, the three-dimensional surface structure plays a fundamental role that various methods are expected to modulate in a number of ways, one most promising of which is the use of different types of radiation. In the present manuscript, we demonstrate in a calvarial defect model, that treatment with ultraviolet irradiation allows modification of the osteoconductive characteristics in a biomaterial formed by gelatin and chitosan, together with the inclusion of hydroxyapatite and titanium oxide nanoparticles.

## 1. Introduction

The development of cell technologies and of materials’ synthesis is the primary goal of tissue engineering with the purpose of restoring, maintaining or improving the function of biological tissues [1]. Particularly, in the treatment of bone injuries, the development of tissue engineering promises to solve the different problems related to the use of conventional treatments, represented by the techniques of autografting, allografting and the use of metal implants [2].

The biomaterials with osteoinductive and osteoconductive characteristics represents one of the most important strategies in the development of therapies for the treatment of osteodegenerative diseases and different bone lesions [3]. Osteoinduction refers to the process of transformation of primitive, undifferentiated and pluripotent cells into lineages that will form new bone [4,5]. This cell differentiation occurs thanks to different environmental factors where hydroxyapatite and titanium nanoparticles are prominent [6,7,8], those promote the transformation of undifferentiated mesenchymal cells into preosteoblasts [9]. On the other side, osteoconduction is a property that establishes the three-dimensional environment favorable for the proliferation of bone cells, depending on the structural framework and surface characteristics of the materials [10]. Several studies have been carried out with the aim of obtaining biocomposites that combine the characteristics of titanium and hydroxyapatite in order to improve the adhesion of osteoblasts, promote proliferation and improve the mechanical characteristics of biomaterials [11,12,13].

Considering osteoconduction and its response to surface modifications of materials, various physical energetic stimuli have been used to generate small structural alterations [14], including the use of different radiation sources [15]. In this sense, modifications of the polymerization processes of materials dependent on radiation and the direct use of radiation on already formed materials stand out [16]. In the latter, the use of ultraviolet radiation, gamma radiation and electron beam irradiation (EBI), have allowed different susceptible materials, to produce surface modifications that improve osteoconduction [17,18,19].

Considering the ability to alter the surface characteristics of titanium nanoparticle composite materials with ultraviolet radiation [19]; here, we report the development of a new biodegradable, ultraviolet-treated nanocomposite material that allows for the enhancement of osteoconduction demonstrated in critical defect calvarial tests.

## 2. Materials and Methods

### 2.1. Membrane Synthesis and Characterization

The synthesis of the membranes was prepared by cold casting, using a method previously reported [20]. Briefly, the chitosan (2%) and nanoparticles were dispersed in acetic acid (1%) (titanium oxide and hydroxyapatite nanoparticles, 500 ppm). This solution was mixed with gelatin (4%) in 3:2 ratio at 50 °C for 1 h, glycerol (1 g per 100 mL of solution) was added. The solution was crosslinked by mix EDC/NHS, to conclude with a cold casting process (3 days at 10 °C). In vitro degradation measurements were performed by measuring percentage weight variations in samples subjected to a lysozyme solution (100 μg/mL) in PBS at 37 °C for 4 weeks. The UV irradiation process was performed using a wavelength of 254 nm (dose: 10 joule/cm^2^) with a commercial UV device (Bio-Link, BLX-254, Vilber Lourmat, France), the exposure time to achieve the dose was approximately 7 min. Membrane microscopy was performed by SEM/EDS (scanning electron microscopy/energy dispersive X-ray spectroscopy), using Carl-Zeiss SEM (EVO.MA.10) system equipped with EDS (X-Act, Oxford Instrument, Abingdon, Oxfordshire, England). The sterilization process was performed using an ethanol battery for 3 h. Characterizations were performed in triplicate. Statistical analysis was performed using *t*-tests *p* ≤ 0.05.

### 2.2. In Vitro Inflammatory Response Analysis

RAW-264.7 murine macrophage cell line was grown with RPMI-1640 supplemented medium (10% fetal bovine serum, pH 7.4) for 5 days at 37 °C and 5% CO_2_ until 100% confluence (Cell lines were purchased from Merck KgaA, Darmstadt, Germany). Control corresponds to commercial gelatin membrane. For the experiments, 2.5 × 10^5^ cells/well were seeded over the membranes in a 12-well ELISA plate for 24 h, to subsequently extract 1 mL of sample. The concentration of pro-inflammatory cytokines, IL-6 and TNF-α secreted were calculated according to the standard curves of each cytokine using LPS as a positive control, standardized in the commercial ELISA kits (Mouse ELISA MAX™ Deluxe Set, Biolegend^®^, San Diego, CA, USA). Characterizations were performed in triplicate. Statistical analysis was performed using *t*-tests *p* ≤ 0.05.

### 2.3. In Vivo Measurements

All experiments were approved by the Animal Care and Use Institutional Committee of the University of Chile (project agreement 19230-MED-UCH). JAX™NOD-SCID mice were purchased from The Jackson Laboratory (Maine, CT, USA). Sixteen two-month-old female or male mice (approximatively 20 g) were used for analysis. Chirurgical procedures were made in SPF conditions. Mice were anesthetized by intraperitoneal injection of Ketamine (100 mg/Kg)/Xylazine (10 mg/Kg). Two symmetrical full-thickness critical cranial defects (3.5 mm diameter) where performed with a tissue punch rotatory drill. The defects were made on each parietal region in the calvaria without touching the skull suture and preserving the dura mater tissue. The implanted membranes were cut circularly with a diameter equivalent to each defect. Animals were randomly separated into two experimental groups consisting of Group 1 (8 animals): one defect was filled with unradiated membrane and the other defect with radiated membrane. Group 2, negative control, (eight animals): one defect empty, the other was filled with unradiated membrane (four animals) or radiated membrane (four animals). Eight mice (four from each group) were sacrificed at the end of the first month and the other eight at 2 months post-surgery. The calvaria region was isolated and fixed for 48 h in 4% paraformaldehyde at 4 °C. For demineralization, hard tissues were submerged in 10% EDTA tamponed with PBS 1×, pH 7.4. The reaction was accelerated with microwaves. Subsequently, the demineralized calvaria were dehydrated in sequential alcohol baths and cleared in xylene to be embedded in paraffin. Serial frontal sections of 5 µm were obtained with a rotatory microtome. Sectioned samples were deparaffinized and stained with Haematoxylin and Eosin methods. Image acquisition was performed using a Leica Zeiss Axio Lab A1 microscope (Wetzlar, Germany), connected with a digital camera Canon EOS Rebel-T3 associated to EOS Utility software [21].

For bone regeneration exploration, were used an X-ray micro-CT device (SkyScan 1278, Bruker, Kontich, Belgium). Four- and eight weeks post-surgery samples were analyzed with the X-ray source set at 65 kV and 701 μA. An internal density phantom, was used to scale bone density. Tissue samples were also examined with scanning electron microscopy in back-scattered electron mode (BSE-SEM) using a JEOL (Tokyo, Japan) IT300LV microscope equipped with X-ray dispersive energy elemental microanalysis (EDX). Previously, the bone samples were fixed in 10 wt% formaldehyde for 48 h, immersed in ascending concentrations of alcohol, and dried in an Autosamdri-815 supercritical CO_2_ dryer [21].

## 3. Results and Discussion

The developed nanocomposite corresponds to a membrane composed of gelatin/chitosan, plus titanium oxide (TiO_2_) nanoparticles and hydroxyapatite (HAp) nanoparticles, which are uniformly distributed in the material with an approximate titanium/calcium ratio 7:3, calculated by SEM/EDX analysis (Figure 1A). The final conformations of the materials included the treatment by ultraviolet radiation of 10 Joule/cm^2^, resulting in material with microstructure different from the control without irradiation (Figure 1C,D). Due to the known electronic interaction of TiO_2_ with UV radiation, energy dissipation is allowed which leads to the modification of the structures, which is evidenced in the SEM cross-sectional analysis of the membranes [22]. Although the treatment with UV radiation often increase the surface hydrophilicity of the material and thus increases the access to the interaction with the aqueous medium [23], the modification of the internal structure produces an increase in Young’s modulus, which we have previously characterized, which is related to a reduction of the degradation in this material [20]. As a result, treatment with UV radiation produces a slow degradation of the material, with significant differences from the first week of analysis (Figure 1B) and a possible increase in the interaction capabilities with bone cells, which is ideal for use as a therapeutic alternative [24]. These results are related to the surface and biological characterizations performed on this membrane, which allowed establishing the correct biological interaction in vitro [20].

Prior to in vivo characterization, the membranes were analyzed to determine the in-flammatory response in vitro and thus establish the safety of the animal tests (Figure 2). The assays measured the expression of TNF-α and interleukin-6 in RAW 264.7 macrophage cells, grown on membranes samples [25]. The results demonstrated a non-significant inflammatory response for all membranes. These results correlate with the low immune response produced by materials that are composed of chitosan, which is known for its ability to modulate the inflammatory response [26].

The in vivo evaluation was carried out by means of trials in a critical calvarial defect model, which allows the repair in an orthotopic bone site, with physiologically relevant results [27]. In this methodology the controlled damage in the bone does not allow spontaneous regeneration, therefore, only the therapeutic intervention allows bone repair [28].

In the histological analysis after 4 weeks post implantation (Figure 3), it is observed the absence of tissue regeneration in the negative control (defect without therapeutic treatment), where the wound manages to close with fibrous tissue delimited by host bone tissue (black line Figure 3A), which represents a normal histological development for this model without treatment [29]. In the defects that were treated with the membranes, it can be seen that the material continues inside the defect, but there is evidence of a displacement of this, losing close contact with the surrounding bone tissue (Figure 3B,E). When this contact is lost, there is a tendency to the formation of fibrous tissue around the membrane that keeps them attached to the ends of the defect, which is normally due to the lack of support for cell migration and the activation of the repair process mediated by inflammatory cells [30]. Where contact was maintained with the surrounding bone tis-sue, new bone tissue growth is observed on both membranes, which advanced from the edges (white line) to the center of the defect.

The neoformed tissue on the membranes is mature bone, with living cells inside, which has not been remodeled. At the same time, blood vessels (white circle) are seen in the vicinity of the material, which also remains in close relationship with the underlying brain tissue (red lines). Histological analysis does not show any difference in the cellular behavior between the two types of membranes.

The analysis by means of micro-CT was used to study the formation of the mineralized matrix of the bone tissue that has migrated towards the treated defects, one and two months after implantation (Figure 4). In all cases the membranes are shown to be radio-opaque ( Figure 4B,E), the colorimetric analysis shows a low uniform density of membrane; however, at the second month it is possible to observe an increase in the density of the membranes, similar to the native bone Figure 4C,F). Membrane density analysis allows us to establish a criterion for the regeneration of bone tissue with the formation of a mineralized matrix, which indicates the efficiency of each treatment [31]. However, this analysis does not allow us to establish differences between the membranes with and without ultraviolet radiation treatment. However, this analysis does not allow us to establish differences between the membranes with and without ultraviolet radiation treatment.

To determine the efficiency of treatments in calvarial defects, it is necessary to estimate, in materials with bone tissue, the bone mineral content (BMC) [31], which is usually calculated by approximating the hydroxyapatite content (HA). However, calculation by elemental analysis (EDX) allows better results to be obtained by considering biomaterials that have HA within their formulation [32]. In Figure 5 it is possible to observe the BSE-SEM/EDX analysis to defects after two months post-implantation. BS-SEM images (Figure 5A–C) show the mineralized areas as brighter zones, while soft tissues are seen in darker color. It can be observed a higher mineralization density on the zone occupied by the membranes as compared to the untreated calvarial defect, particularly on the UV–irradiated membrane.

These results are consistent with the density and distribution of elements. This has been directly related to the efficiency of the regenerative process [33,34,35,36]. Is also shown the negative control with organic/fibrous tissue rich in carbon and oxygen. Native white bone is observed with similar patterns in some portions of the membranes (Figure 5B,C). For all the samples, calcium deposits (small red dots, Figure 5G–I), while in the case of the membranes (Figure 5E,F) a higher concentration of calcium is also observed in the peripheries of the defect in direct relation to the native bone and the cellular migration from it. This phenomenon is more evident in the radiated membrane with a significant increase in the concentration of both calcium and phosphorus, according to the quantification of elements (Figure 5J–L).

The elemental measurement also allows to establish the degradation speed of the membranes when measuring the reduction of the carbon and oxygen concentration in the samples, in this sense the differences with the in vitro models are due to the location of the defects, where there is less capacity of enzymatic systems of degradation [37], along with the greater enzymatic stability derived from the presence of HA [38].

Osteoconduction is a process highly dependent on the conditions provided by the materials used in bone tissue engineering and driven by the process of cell proliferation, mediated or not by the incorporation of bioactive elements [5]. While the osteoconductive properties of the elements present in the membranes used in the present study are recognized [39], the differences of components of the mineralization process provides evidence of desirable performance in the regenerative process [40,41,42], which is related to the irradiated membranes and probably to the related microstructural changes [43]. In general, modulation of regenerative responses and osteoconduction can be achieved by subjecting materials to radiation exposures of different nature [35], the susceptibility to the generation of structural changes will depend on the characteristics of the material and the radiation source which establishes an area of research that deserves further attention.

## 4. Conclusions

In the present investigation, a material made of titanium oxide nanoparticles can have its structural properties modified by treatment with ultraviolet radiation. These modifications suggest a change in the performance of the material as a scaffold in bone tissue regeneration and the consequent osteoconductive property. The results shown provide a prominent background for ultraviolet treatment to be consistently studied for the modulation of regenerative properties and thus be a fundamental part of the development of future therapeutic alternatives.

## Figures and Tables

**Figure 1 polymers-14-00289-f001:**
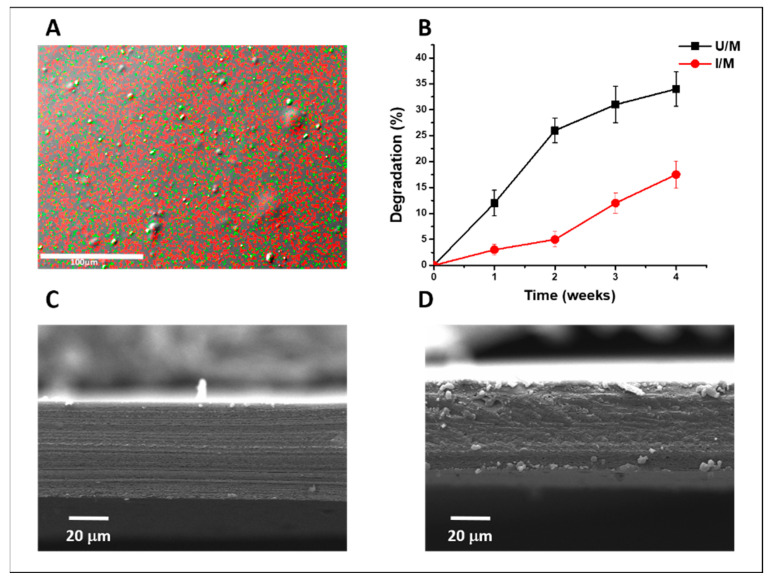
Membranes characterizations. (**A**) Distribution of nanoparticles in gelatin/chitosan membrane by SEM/EDS analysis, calcium (green dots) and titanium (red dots). (**B**) enzymatic degradation assay. (**C**,**D**) cross-section membrane microstructure, SEM microphotographs. U/M: membrane non-UV-irradiated, I/M: membranes UV-irradiated.

**Figure 2 polymers-14-00289-f002:**
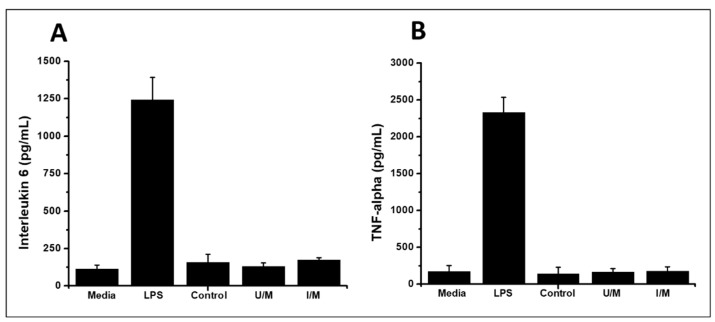
Expression of inflammatory molecules. (**A**) Expression of interleukin-6, (**B**) expression of TNF-α.

**Figure 3 polymers-14-00289-f003:**
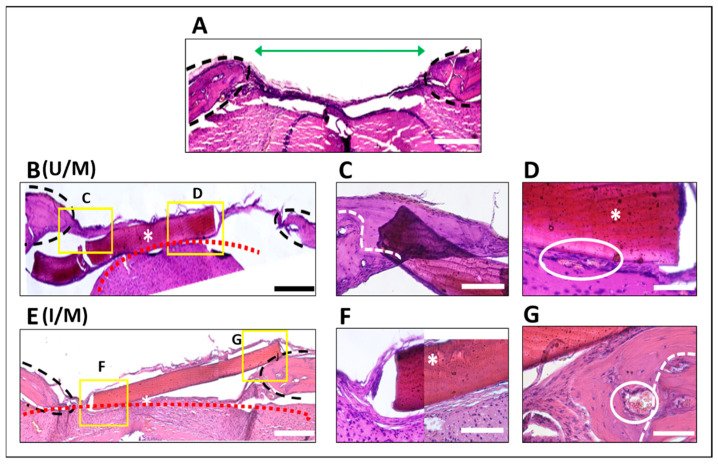
Histological analysis in calvarial defect model. (**A**) Negative control (untreated calvarial defect) green arrow indicates the area of the defect. (**B**,**E**) correspond to the representative images of the defects treated with membranes unirradiated (U/M) and UV irradiated respectively (I/M). Amplifications of the zones are shown in yellow squares for figure (**B**) (U/M) in (**C**,**D**). Amplifications of the yellow squares for figure C (I/M) are shown at (**F**,**G**). The white asterisks indicate membranes. (Figure (**A**,**B**,**E**) 4× magnification, 1000 µm: Figure (**C**,**D**,**F**,**G**) 40× magnification, scale bar 500 µm).

**Figure 4 polymers-14-00289-f004:**
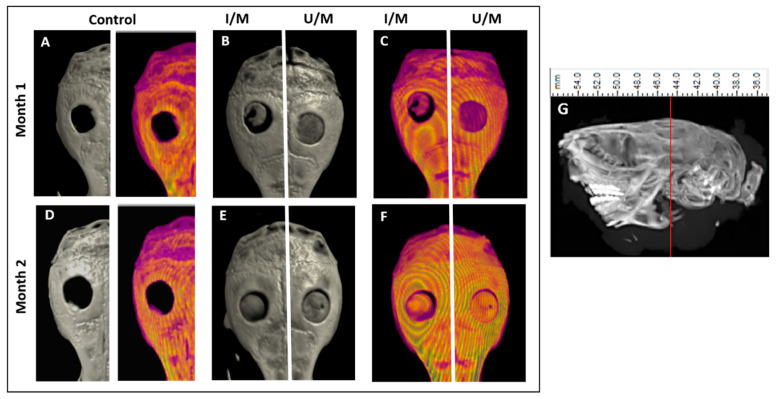
Micro-CT analysis in calvarial defect model. (**A**–**F**) Top view of the coronal plane of the mouse skull. In (**A**,**D**) negative controls (untreated calvarial defect). In (**B**,**C**,**E**,**F**) the membranes inside the critical defects in the cranial vault are observed. (**G**) Side plan and dimensions.

**Figure 5 polymers-14-00289-f005:**
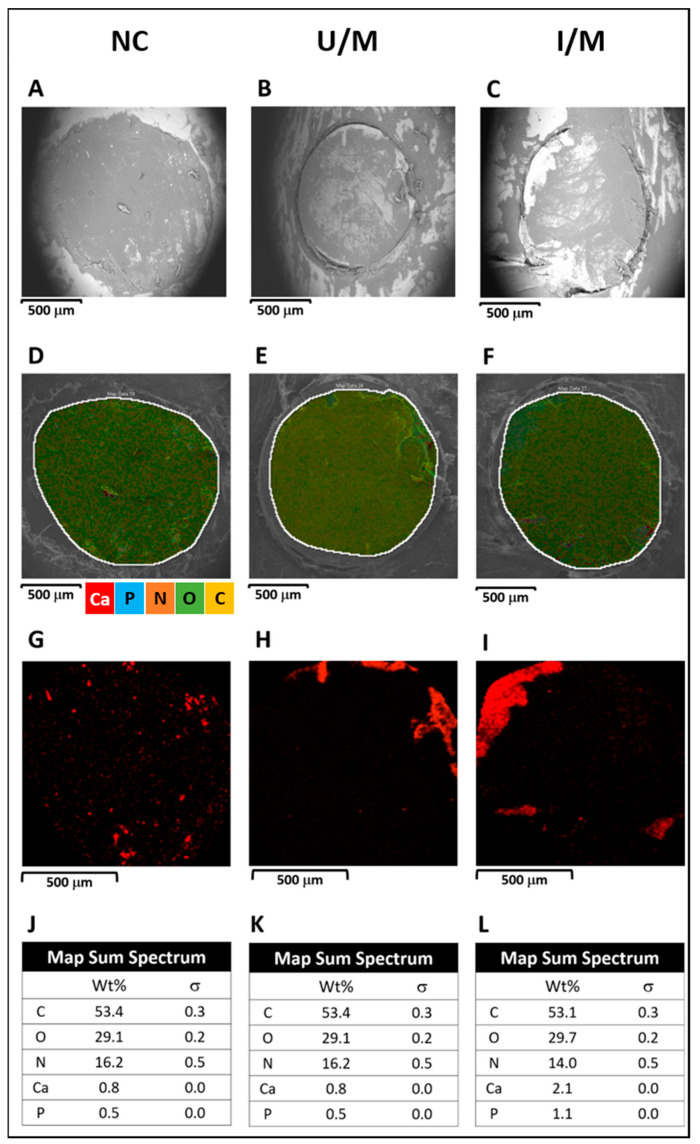
BS-SEM images (**A**–**C**), multi-element EDX mapping (**D**–**F**), calcium EDX mapping (**G**–**I**), and EDX element contents (**J**–**L**). Each images columns corresponds to: NC (negative control, untreated calvarial defect), U/M: membrane non-UV-irradiated and I/M: membranes UV-irradiated.

## Data Availability

Not applicable.

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
