# Peer review of "Osteoconductive Effect of a Nanocomposite Membrane Treated with UV Radiation"

_polymers, 2022, doi:10.3390/polym14020289_

Round 1

Reviewer 1 Report

Authors claim that a treatment of the calvarial defect with ultraviolet irradiation allows modifying the osteoconductive characteristics in a biomaterial formed by gelatin and chitosan, together with the inclusion of hydroxyapatite and titanium oxide nanoparticles.

Generally, the methods are clear, scientific, and accessible. The results are clearly presented. Overall the manuscript has merit, but not reaches the published level for Polymers. Some input may help.

  1. The concepts of osteconductive and osteoinductive are relatively simple. One can define that osteoconduction refers to a bone graft's ability to provide a structural framework on which host cells reconstitute. Osteoinduction is a differentiation of stem cells into osteoblasts, resulting in growing into the graft site. Please address and define in detail.
  2. The UV radiation is the method. There is no any description in the introduction. Please address and explain why select UV.
  3. Why select UV dose of 254 nm? Please define this dose, and time period.
  4. Gamma ray irradiation has been used in the bone graft material to increase osteoconduction/osteoinduction. Please compare the Gamma ray effect with UV effect in the introduction and discussion.

Author Response

Thank you very much for your comments and suggestions. In this regard;

1. The concepts of osteconductive and osteoinductive are relatively simple. One can define that osteoconduction refers to a bone graft's ability to provide a structural framework on which host cells reconstitute. Osteoinduction is a differentiation of stem cells into osteoblasts, resulting in growing into the graft site. Please address and define in detail.

Suggested modifications were introduced, incorporating new bibliographies and appropriate definitions in the introduction section.

2. The UV radiation is the method. There is no any description in the introduction. Please address and explain why select UV.

Your suggestion has been considered and the background information is incorporated in the introduction. 

3. Why select UV dose of 254 nm? Please define this dose, and time period.

Modifications were made to materials and methods. The exposure time is automatically calculated by the equipment to reach the radiation dose, considering the radiation source with 40 Watt lamps and the dimensions of the treatment booth.

4. Gamma ray irradiation has been used in the bone graft material to increase osteoconduction/osteoinduction. Please compare the Gamma ray effect with UV effect in the introduction and discussion.

We appreciate this suggestion and include it in the introduction, discussion and Conclusions of the manuscript. 

Reviewer 2 Report

Dear authors: 

The study entitled" Osteoconductive effect of a nanocomposite membrane treated with UV radiation" shows an interesting approach to the application of membranes for bone regeneration. The authors proposed the incorporation of HA and Ti nanoparticles to your membrane; and an additional UV treatment. Moreover, the authors explored some biological responses using in vitro methods and in vivo methods. However, the characterization presented for this new biomaterial is very simple and require intense improvement. In my opinion, this study should be original research for receiving your desired merit. 

Some substantial suggestions for improvement: 

1. The introduction section is too short. The authors need to explore more the recent articles proposing similar methodologies and expose the problem/justification more in-deep.

2. The characterization using SEM/EDS is too simple to prove that the authors really included nanoparticles into the membrane. Did the authors map by EDS the membrane previous to the biological assays? Where is the previous quantification? Ca.. P...  

3. The in vitro tests showed responses only for 24h. I think the authors need to increase this time by at least one week. In order to prove if the Ti nanoparticle can affect or not cells. Some current reports are showing the possibility of cytotoxicity provide by nanoparticles release. This is a point that deserves discussion.

4. At any point in this study I saw the detailing of statistics or sample numbers. At least, in the characterization and in vitro methods. This needs improvement. 

5. The authors reported (line 198 and Figure 5) that the level of Ca/P increased significantly for the irradiated membrane. In my opinion, is necessary some statistics proving this "significant improvement". The upgrade from (0.8 to 2.1) for Ca and from (0.5 to 1.1) for P, seems not as a significant improvement.

6. The UV-treatment is not clear. Times applied? Distance? Storage of the sample? Sterilization process to in vitro and in vivo tests.

7. At this moment, I can't agree with the conclusions section. None of the tests proved completely the incorporation of Ti nanoparticles in the characterization section. In vitro tests were performed only for 24h and can not be a basement that the molecular events are perfect for this biomaterial with UV treatment. The study needs to explain your limitations.

Author Response

Thank you very much for your comments and suggestions. In this regard;

  1. The introduction section is too short. The authors need to explore more the recent articles proposing similar methodologies and expose the problem/justification more in-deep.

The introduction has been modified for a better explanation, trying to respect the instructions of the magazine for short communications.

  1. The characterization using SEM/EDS is too simple to prove that the authors really included nanoparticles into the membrane. Did the authors map by EDS the membrane previous to the biological assays? Where is the previous quantification? Ca.. P...

The elemental analysis performed as part of the material characterization (SEM/EDS) was carried out prior to the biological tests and was intended to determine the presence and distribution of the calcium and titanium nanoparticles included in the biomaterial synthesis process. In this analysis, the normalized ratio of the emitted signals is presented in the results section.

The following elemental analysis was performed together with the MicroCT technique on the materials implanted in the in vivo test. This analysis looks for the presence of elements related to new bone formation.  3. The in vitro tests showed responses only for 24h. I think the authors need to increase this time by at least one week. In order to prove if the Ti nanoparticle can affect or not cells. Some current reports are showing the possibility of cytotoxicity provide by nanoparticles release. This is a point that deserves discussion.

  1. At any point in this study I saw the detailing of statistics or sample numbers. At least, in the characterization and in vitro methods. This needs improvement.

We appreciate this comment. The number of samples are indicated in materials and methods.

  1. The authors reported (line 198 and Figure 5) that the level of Ca/P increased significantly for the irradiated membrane. In my opinion, is necessary some statistics proving this "significant improvement". The upgrade from (0.8 to 2.1) for Ca and from (0.5 to 1.1) for P, seems not as a significant improvement.

We understand this statistical concern, which is derived from the limitations imposed by the bioethics committee on the number of animals used. However, under our experience and comparative results with other investigations, it allows us to have a consistent level of security for our deductions, however, we understand the doubts that may occur, therefore we have modified the discussion by adding an explanation with appropriate citations, in addition, we have moderated the discussion of results so that it is understood that our results "suggest" a mineralization process.

  1. The UV-treatment is not clear. Times applied? Distance? Storage of the sample? Sterilization process to in vitro and in vivo tests.

We have modified the explanation of the methodology for understanding radiation doses in UV treatment. We have also included the methodology for sterilization of the materials.

  1. At this moment, I can't agree with the conclusions section. None of the tests proved completely the incorporation of Ti nanoparticles in the characterization section. In vitro tests were performed only for 24h and can not be a basement that the molecular events are perfect for this biomaterial with UV treatment. The study needs to explain your limitations.

Normally, compounds performing an anti-inflammatory effect are studied during the first 24 hr. Since, inflammation occurs during this time. Our model is one of the most used model to validate anti-inflammatory effects .

We understand the attached comments and have modified the conclusions and discussion to explain the limitations of our results. 

Reviewer 3 Report

  1. It will be better to provide the UV-treatment time in the membrane synthesis and characterization section of the manuscript.
  2. Please mention the source for RAW 264.7 cells in the manuscript.
  3. What was the weight of the polymer films used in this study for in vivo analysis?
  4. It is recommended to measure the surface roughness and hydrophilicity potential of the treated samples.
  5. Please mention the sample concentrations used in the analysis of interleukin and TNF-a in the manuscript.
  6. 5. Please mention the scale bar in Figures 3 and 4.
  7. It is recommended to measure the other important parameters related to bone formation, such as bone volume fraction, bone surface density, etc.

Author Response

Thank you very much for your comments and suggestions. In this regard;

  1. It will be better to provide the UV-treatment time in the membrane synthesis and characterization section of the manuscript.

Modifications were made to materials and methods. The exposure time is automatically calculated by the equipment to reach the radiation dose, considering the radiation source with 40 Watt lamps and the dimensions of the treatment booth.

  1. Please mention the source for RAW 264.7 cells in the manuscript.

RAW 264.7 cells were purchased from Merck. It was added in the materials and methods section 

  1. What was the weight of the polymer films used in this study for in vivo analysis?

The dry weight of the membranes of 1 cm2 is approximately 0.1 g.

  1. It is recommended to measure the surface roughness and hydrophilicity potential of the treated samples.

Hydrophobicity is a parameter measured in the first part of this research which has already been published. Comments have been added in the discussion that allude to these measurements.

  1. Please mention the sample concentrations used in the analysis of interleukin and TNF-a in the manuscript.

The amount of sample used from the supernatant of each culture is 100 uL. The data is added in the experimental section

  1. Please mention the scale bar in Figures 3 and 4.

The figures shown have been modified according to your indications.

  1. It is recommended to measure the other important parameters related to bone formation, such as bone volume fraction, bone surface density, etc.

Bone formation parameters such as density are analyzed semi-quantitatively in the discussion of histological and MicroCT results.

Reviewer 4 Report

The reading of the manuscript “Osteoconductive effect of a nanocomposite membrane treated with UV radiation” is difficult and confusing. The authors have dealt with the subject superficially without even pointing out why the UV radiation has been chosen to improve the osteoconducive properties of biomaterials. How UV irradiation improves the osteoconducive property of any biomaterials? What is the mechanism behind it?

The Materials & Methods section is incomplete. The manuscript further did not mention the duration of UV irradiation? What is the effect of too long or too short exposure of UV irradiation on osteoconducive properties of biomaterials? What is the most appropriate treatment condition (UV)? Without this information, it is not possible to reproduce the results.

The authors are too much optimistic with UV irradiated samples. There is not significant difference between UV treated and untreated samples in Micro-CT analysis in calvarial defect model [Figure 4].

Eventually I reached the conclusion that in its present form this manuscript resembles more a draft than a finished and well-polished work. The manuscript in its present form citing a number of problems, including, but not limited to, poorly constructed English-use (highly complex sentences), unclear segments and the failure to critically evaluate the subject. This ends up fatiguing and bewildering rather than enticing the reader.

In addition, all figures should be labelled uniformly. Figure 3 should be labeled as Figure 4 (I/M, U/M ets).

Author Response

Thank you very much for your comments and suggestions. In this regard;

  1. The reading of the manuscript “Osteoconductive effect of a nanocomposite membrane treated with UV radiation” is difficult and confusing. The authors have dealt with the subject superficially without even pointing out why the UV radiation has been chosen to improve the osteoconducive properties of biomaterials. How UV irradiation improves the osteoconducive property of any biomaterials? What is the mechanism behind it?

Thank you very much for your comment. We have consistently improved the wording of the manuscript by referring to the issues you raised in the introduction and discussion.

  1. The Materials & Methods section is incomplete. The manuscript further did not mention the duration of UV irradiation? What is the effect of too long or too short exposure of UV irradiation on osteoconducive properties of biomaterials? What is the most appropriate treatment condition (UV)? Without this information, it is not possible to reproduce the results.

Thank you very much for your comment. We have modified the materials and methods section. The exposure time is automatically calculated by the equipment to reach the radiation dose, considering the radiation source with 40 Watt lamps and the dimensions of the treatment booth.

Other characterizations are part of the first part of this research which is already published.

  1. The authors are too much optimistic with UV irradiated samples. There is not significant difference between UV treated and untreated samples in Micro-CT analysis in calvarial defect model [Figure 4].

Thank you very much for your comment. We believe the process of mineralization to be consistent as an indication of the regeneration process, however, we have modified our wording so that further research will provide more background to make the conclusions drawn in this manuscript more robust.

  1. Eventually I reached the conclusion that in its present form this manuscript resembles more a draft than a finished and well-polished work. The manuscript in its present form citing a number of problems, including, but not limited to, poorly constructed English-use (highly complex sentences), unclear segments and the failure to critically evaluate the subject. This ends up fatiguing and bewildering rather than enticing the reader.

We have modified the wording of the manuscript in the light of your comments.

In addition, all figures should be labelled uniformly. Figure 3 should be labeled as Figure 4 (I/M, U/M ets).

The figures were modified according to your comment 

Round 2

Reviewer 2 Report

The authors corrected important content in the manuscript and the conclusions are in accordance with the study. However, the manuscript continues with some lack of information. 

Cite that the experiment was made in triplicate did not give correct statistical information for the readers. How do I know that the authors found or did not have statistical significance in the cellular experimentations or in degradation tests? Which statistical methods were applied? Software? Significance? 

Author Response

Thank you very much for your comments; in methods we have added the statistical test used for the comparison of variances and in the results section we have added comments on Figure 1B.

Reviewer 4 Report

The revisions that authors are made to the manuscript are very effective in addressing the concerns previously raised.

Author Response

Thank you very much for your comments; we have added some additional improvements as requested. 

Round 3

Reviewer 2 Report

The authors tried to respond to all the issues proposed. No more questions.